

# Drivers of atmospheric methane uptake by montane forest soils in the southern Peruvian Andes

S. P. Jones[†1], T. Diem[†2], L. P. Huaraca Quispe[3], A. J. Cahuana[3], D. S. Reay[1], P. Meir[1,4] and Y. A. Teh[2]

[1]School of Geosciences, University of Edinburgh, Edinburgh, UK
[2]Institute of Biological and Environmental Sciences, University of Aberdeen, Aberdeen, UK
[3]Universidad Nacional de San Antonio Abad del Cusco, Cusco, Peru
[4]Research School of Biology, Australian National University, Canberra, Australia

[†]contributed equally to the work

*Correspondence to*: Sam Jones (sjones@bordeaux.inra.fr)

**Abstract.** The soils of tropical montane forests can act as sources or sinks of atmospheric methane ($CH_4$). Understanding this activity is important in regional atmospheric $CH_4$ budgets, given that these ecosystems account for substantial portions of the landscape in mountainous areas like the Andes. Here we investigate the drivers of $CH_4$ fluxes from premontane, lower and upper montane forests, experiencing a seasonal climate, in southeastern Peru. Between February 2011 and June 2013, these soils all functioned as net sinks for atmospheric $CH_4$. Mean (standard error) net $CH_4$ fluxes for the dry and wet season were -1.6 (0.1) and -1.1 (0.1) mg $CH_4$ - C m$^{-2}$ d$^{-1}$ in the upper montane forest; -1.1 (0.1) and -1.0 (0.1) mg $CH_4$ - C m$^{-2}$ d$^{-1}$ in the lower montane forest; and -0.2 (0.1) and -0.1 (0.1) mg $CH_4$ - C m$^{-2}$ d$^{-1}$ in the premontane forest. Variations among forest types were best explained by available nitrate and water-filled pore space, indicating that nitrate inhibition of oxidation or diffusional constraints imposed by changes in water-filled pore space on methanotrophic communities represent important controls on soil-atmosphere $CH_4$ exchange. Seasonality in $CH_4$ exchange varied among forests with an increase in wet season net $CH_4$ flux only apparent in the upper montane forest. Net $CH_4$ flux was inversely related to elevation; a pattern that differs to that observed in Ecuador, the only other extant study site of soil-atmosphere $CH_4$ exchange in the tropical Andes. This may result from differences in rainfall patterns between the regions, suggesting that attention should be paid to the role of rainfall and soil moisture dynamics in modulating $CH_4$ uptake by the organic-rich soils typical of high elevation tropical forests.

## 1 Introduction

Methane ($CH_4$) is an important greenhouse gas, accounting for at least a fifth of the climate forcing associated with increases in the atmospheric concentration of well-mixed greenhouse gases since the industrial revolution (Cicerone and Oremland, 1988; Myhre et al., 2013). Despite the importance of tropical landscapes in the global $CH_4$ budget, the comparison of satellite retrievals of the atmospheric concentration of $CH_4$ with source-sink inventories and bottom-up process based models



indicates that these landscapes are poorly characterised (Bergamaschi et al., 2009; Bloom et al., 2010; Frankenberg et al., 2005). This likely reflects a historic imbalance in field observations, when compared to the northern hemisphere. Soils play a key role in controlling atmospheric $CH_4$ concentrations, with emissions from inundated tropical wetland soils representing the largest natural source of atmospheric $CH_4$, whilst well-drained soils represent the largest net biological sink (Ciais et al., 2013). Soils, being capable of acting as both globally significant sources or sinks for atmospheric $CH_4$, are of particular interest in refining our understanding of $CH_4$ exchange across tropical landscapes (Dutaur and Verchot, 2007; Spahni et al., 2011).

The function of soil as source or sink for atmospheric $CH_4$ is the net result of consumption and production by aerobic methanotrophic bacteria and anaerobic methanogenic archaea respectively (Conrad, 1996; Le Mer and Roger, 2001). Well-drained soils are typically thought to act as a net sink for atmospheric $CH_4$ because aerated soils support communities of high-affinity methanotrophic bacteria that oxidise $CH_4$ at near-ambient concentrations (Bender and Conrad, 1992; Teh et al., 2006). In such soils, variations in the net flux of $CH_4$ between soil and atmosphere are expected to be strongly influenced by constraints on the diffusional supply of $CH_4$ to methanotrophs imposed by the structure of the soil pore network and the difference between gaseous and aqueous phase mass transfer of $CH_4$ (Bender and Conrad, 1992; Smith et al., 2003). This reliance on diffusion indicates that high-affinity methanotrophs are likely to occupy well-connected pore spaces, and as such uptake of atmospheric $CH_4$ is also sensitive to water limitation under drier conditions (von Fischer et al., 2009). Additionally, methanotrophic activity may be inhibited by the presence of inorganic nitrogen; for example, through competition between $CH_4$ and ammonium ($NH_4^+$) for the active sites of the enzyme facilitating oxidation (Reay and Nedwell, 2004; Steudler et al., 1989). However, well-drained soils may also support anaerobic processes, concurrent with oxic conditions in the bulk soil matrix, within anoxic microsites (Conrad, 1996; Sexstone et al., 1985; Teh et al., 2005). These anoxic microsites form as a result of physical limitations on the rate of $O_2$ diffusion imposed by aggregate structure or saturation of soil pores both of which may promote the development of radial $O_2$ gradients around these occluded microsites (Burgin et al., 2011; Sexstone et al., 1985). Where biological $O_2$ demand outstrips diffusional supply these gradients can result in localised anoxia (Burgin et al., 2011; Verchot et al., 2000). Under wet conditions and high $O_2$ demand, anaerobic metabolic activity can be significant and potentially lead to $CH_4$ emissions (Silver et al., 1999; Teh et al., 2005; Verchot et al., 2000). In these environments, methanotrophy can consume the majority of $CH_4$ produced in-situ through the activity of low affinity communities utilising elevated $CH_4$ concentrations at the interface of anoxic and oxic zones (Conrad, 1996; Teh et al., 2005). Consequently, variations in net flux may be expected to result from changes in relative size and connectivity of anoxic versus oxic zones, or from competition for substrates among methanogens and other anaerobes using more energetically favourable metabolic pathways such as the reduction of nitrate ($NO_3^-$) (Chidthaisong and Conrad, 2000; von Fischer and Hedin, 2007; Silver et al., 1999; Teh et al., 2008).



Well-drained tropical soils are estimated to account for approximately a third of the global atmospheric soil sink for $CH_4$ with nearly three-quarters of this uptake occurring within forest environments (Dutaur and Verchot, 2007). In Central and South America, tropical forests are expansive covering ∼35 % of the South American continent, and exhibiting considerable spatial and temporal variability in soil-atmosphere $CH_4$ exchange (Davidson et al., 2004; Eva et al., 2004; Keller et al., 1986, 2005; Steudler et al., 1996; Verchot et al., 2000). The majority of studies in the region have focussed on lowland forests below 600 m asl. Variations in the strength of the soils of these forests as a sink for atmospheric $CH_4$ are typically explained by the influence of soil texture and moisture on diffusion of $CH_4$ belowground. Whilst $CH_4$ emissions have been linked to the development of anoxic microsites in soils due to high levels of aerobic respiration, combined with periods of high water content (Dutaur and Verchot, 2007; Kiese et al., 2008; Verchot et al., 2000). Tropical montane forests are a spatially extensive component of the tropical forest of South America, accounting for ∼8 % of continental and ∼25 % of Andean landcover, and yet they are under-represented in atmospheric $CH_4$ budgets (Eva et al., 2004; Tovar et al., 2013). Currently, observations in the tropical Americas have been limited to Ecuador (Wolf et al., 2012), Brazil (Sousa Neto et al., 2011), Puerto Rico (Silver et al., 1999; Teh et al., 2005) and Panama (Veldkamp et al., 2013), making it difficult to predict the role of these environments in the regional $CH_4$ cycle.

Moreover, it is possible that controls on $CH_4$ flux from tropical montane forests may differ from their lowland forest counterparts. While the majority of tropical montane forests act as a net sink for atmospheric $CH_4$ (Ishizuka et al., 2005b; Kiese et al., 2008; Purbopuspito et al., 2006; Sousa Neto et al., 2011; Veldkamp et al., 2013; Werner et al., 2006, 2007; Wolf et al., 2012), some ecosystems function as a net atmospheric source, or fluctuate between source and sink (Delmas et al., 1992; von Fischer and Hedin, 2007; Schuur et al., 2001; Silver et al., 1999; Teh et al., 2005). Differences in behaviour among these environments may be partially explained by differences in underlying soil properties. The upper soil horizons of tropical montane forests typically accumulate more organic material than their lowland counterparts, leading to significant differences in the availability of labile carbon and nitrogen, and the evolution of very different soil structure (Nottingham et al., 2012; Zimmermann et al., 2009a). Similar contrasts exist between tropical montane forests where thick organic horizons develop (Purbopuspito et al., 2006) and those where the superficial soils are organo-mineral in origin (Silver et al., 1999). For example, in Ecuador, net $CH_4$ uptake across a tropical montane forest altitudinal transect was better predicted by $CO_2$ flux, ammonium concentration and pH, than soil moisture and texture (Wolf et al., 2012). Likewise, in analogous studies in Indonesia, Brazil and Northern Australia, variations in soil moisture content over time had little or no effect on net $CH_4$ uptake (Kiese et al., 2008; Purbopuspito et al., 2006; Sousa Neto et al., 2011). This is significant given that soil moisture and texture are typically strong predictors of net $CH_4$ uptake in lowland ecosystems (Verchot et al., 2000), and play an important role in mechanistic models of soil $CH_4$ uptake (Curry, 2007). In this context, evidence of nitrogen limitation of $CH_4$ uptake in both lowland and montane tropical forests (Veldkamp et al., 2013; Wolf et al., 2012) or evidence that $CH_4$ production, driven by variations in soil $O_2$ concentration, can play a significant role in the $CH_4$ cycle of some tropical montane forests may help to explain these discrepancies (von Fischer and Hedin, 2007; Silver et al., 1999; Teh et al., 2005).



Here we present a study of soil-atmosphere $CH_4$ exchange, for the period February 2011 to June 2013, from Andean upper montane, lower montane and premontane forests in south-eastern Peru that experience seasonal precipitation. A preliminary short-term dataset indicated that these forests act as a seasonably-variable sink for atmospheric $CH_4$ and that differences in net $CH_4$ flux across the transition from forest to high-altitude grassland is driven by decreases in soil $O_2$ concentration (Teh et al., 2014). Here we aim to: 1) provide an assessment of variations in soil-atmosphere $CH_4$ exchange among and within forest types across this landscape, based on a longer time series; and 2) investigate the drivers of $CH_4$ flux among and within forest types.

## 2 Materials and methods

### 2.1 Study sites

Three sites at 1070 - 1088, 1532 - 1768 and 2811 - 2962 m above sea level (asl), in the south-eastern Peruvian department of Cusco, were selected as they represent montane and premontane forests typical of the eastern flank of the Andes. In this region premontane forests extend from 600 to 1200 m asl, lower montane cloud forests from 1200 to 2200 m asl and upper montane cloud forests from 2200 m asl to the tree line at 3400 m asl (Clark et al., 2014; Zimmermann et al., 2010a). The sites at 2811 - 2962 and 1532 - 1768 m asl representing upper and lower montane cloud forest, respectively, were situated close to the long term study sites of the Andes Biodiversity and Ecosystems Research Group (ABERG) altitudinal transect (Malhi et al., 2010).  A new site was established in premontane forest at 1070 – 1088 because of difficult access at the original ABERG site at this elevation in 2010 - 2011. Site characteristics are summarised in Table 1.

The regional climate is seasonal with decreased rainfall and slightly lower temperatures during the dry season between May and September, with this pattern becoming more pronounced at higher elevation (Fig. 1). Precipitation and air temperature are greater at lower elevations with annual means at the upper and lower montane forest sites of, respectively, 1700 and 2600 mm for precipitation and 12.5 and 18.8 °C for temperature (Girardin et al., 2010). Mean annual precipitation and temperature at the premontane forest site is 5300 mm and 24.4 °C.

The soils of these forests vary with elevation, most notably the surface soils in the montane forests typically consisted of thick organic horizons, ∼ 20 cm deep in the upper and ∼ 10 cm deep in the lower site, whilst, those in the premontane forest were principally mineral in origin (Girardin et al., 2010; Zimmermann et al., 2009a). This pattern is reflected in the carbon contents of these soils with typical values for the upper 10 cm at the upper and lower montane forest sites of 40 - 50 % C and < 5 % C at the premontane forest site (Zimmermann et al., 2009a). These soils are acidic with a pH range of < 4.0.





Soil-atmosphere $CH_4$ exchange for 2011 has previously been reported for these sites as part of a study investigating non-$CO_2$ trace gas fluxes along an Andean altitudinal transect (Teh et al., 2014). These measurements indicate that the $CH_4$ fluxes in the forests are small in comparison to source activity associated with wetlands in the montane grasslands found above the tree-line, with differences in $CH_4$ flux across this gradient best explained by a non-linear inverse relationship with $O_2$

concentration. In this analysis, the montane forests sites acted as sinks for atmospheric $CH_4$, whilst, the premontane forest had the potential to act as both a source or sink.

## 2.2 Sampling strategy

Within each forest type three plots of 20 by 20 m were established approximately three months prior to the start of measurements in an attempt to minimise the effects of disturbances involved with installing sampling equipment (Varner et

al., 2003). Within forest types the distance between plots ranged from ∼ 100 to 1000 m. The plots in the premontane forest were each situated on a ridge, slope and flat feature between elevations of 1070 to 1088 m asl. Similarly, the lower montane forest plots were established on ridge, slope and flat features between elevations of 1532 to 1768 m asl. In the upper montane forest two plots were situated on slopes and the third on a ridge at elevations between 2811 to 2962 m asl.

Within each plot five soil collars were installed to allow for measurements of soil-atmosphere gas exchange using a static flux chamber method. Additionally, soil-gas equilibration chambers were buried at 10 cm adjacent to three collars in each plot to allow measurement of soil $O_2$ concentrations. From September 2011 onwards, five ion exchange resin bags were buried in the upper 10 cm of each plot, adjacent to soil collars, to allow measurement of available inorganic ammonium ($NH_4^+$) and nitrate ($NO_3^-$). Despite the three plots within each forest type broadly occurring within the same forest stand they

were considered independent replicates of forest type as spatial correlations between net $CH_4$ fluxes in tropical forests are small (Ishizuka et al., 2005a; Purbopuspito et al., 2006).

Plots were visited monthly to measure soil-atmosphere gas exchange at each collar, and soil moisture and temperature adjacent to each collar, together with soil $O_2$ concentration in each soil-gas equilibration chamber. Resin bags were also

collected and replaced with new bags during these visits. In the upper and lower montane forests, measurements ran from February 2011 to June 2013. In the premontane forest measurements ran from July 2011 to June 2013. No data are available for the plots of this site in October or December of 2011 and February, July or December of 2012 as high river levels prevented access.

## 2.3 Soil-atmosphere gas exchange

Net soil-atmosphere fluxes of $CH_4$ and $CO_2$, were determined using a static chamber method (Livingston and Hutchinston, 1995). Measurements were initiated by gently sealing cylindrical caps, using a section of inner-tube, to pre-installed soil collars to create a chamber of ∼ 0.08 $m^3$ over a soil surface area of ∼ 0.03 $m^2$. Soil collars had a diameter of 20 cm and were



inserted to a depth of ∼ 5 cm. Each cap was equipped with a gas sampling port, a vent and a 9 V computer fan (Hutchinson and Mosier, 1981; Pumpanen et al., 2004). Using a stopcock and 60 ml gas tight syringe, 20 ml gas samples were taken from the chambers at four discrete time steps over a period of ∼ 30 min. Additionally, air temperature and atmospheric pressure were measured using a type K thermocouple (Omega Engineering Ltd., UK.) and a Garmin GPSmap 60CSx (Garmin Ltd.,USA). Gas samples were stored in over-pressured, pre-evacuated 12 ml Exetainers (Labco Ltd., UK) and concentrations of $CH_4$ and $CO_2$ were determined by gas chromatography. Gas chromatography was conducted using a Thermo TRACE GC Ultra (Thermo Fisher Scientific Inc., USA) with a helium carrier gas at the University of St Andrews. A flame ionization detector (FID) and methanizer-FID were used to determine $CH_4$ and $CO_2$ concentrations, respectively. Analytes were separated using a Hayesep Q 100/200 column. The gas chromatograph was equipped with a 2 ml sample loop and oven temperature was 60 °C. Detector responses were calibrated using three certified gas standards (CK Gas Products Ltd., UK: 1.8, 9.8 and 99.5 ppmv $CH_4$) and instrumental precision was deemed acceptable when coefficient of variances < 5 % were achieved. A custom-built auto-sampler (University of York, UK) was used to introduce gas samples directly into the sample loop.

Fluxes, in $\mu l \ l^{-1} \ m^{-2} \ s^{-1}$ , were calculated in R (R Core Team, 2013) using the HMR package (Pedersen, 2012). Following the criteria outlined by Pedersen et al. (2010), HMR or linear models were fitted to time-series of concentration in chamber headspaces. Significance was determined at the $p < 0.05$ level with emission and uptake indicated by positive and negative flux values, respectively. Non-significant fluxes were excluded from further analysis. Fluxes were converted from a concentration to amount basis reported in mg C $m^{-2} \ d^{-1}$ of $CH_4$ and g C $m^{-2} \ d^{-1}$ of $CO_2$, following the ideal gas law, using measurements of air temperature and ambient pressure.

**2.4 Soil environmental conditions**

Soil $O_2$ concentration was measured from soil gas equilibration chambers buried at 10 cm below the soil surface (Hall et al., 2013; Liptzin et al., 2011; Silver et al., 1999; Teh et al., 2005). Soil $O_2$ concentration was determined by withdrawing 40 ml of gas from a soil-gas equilibration chamber using a stopcock and gas tight syringe. The sample was then passed through the flow-through head of an MO-200 oxygen sensor (Apogee Instruments Inc., USA) into a second syringe. The $O_2$ reading was recorded at a constant volume and the gas sample re-injected into the soil-gas equilibration chamber from the second syringe. Prior to measurements the $O_2$ sensor was calibrated, as required, in the field with ambient air and the dead volumes of the sampling apparatus evacuated to minimise contamination of the soil gas sample by residual atmospheric air. Chambers had an interval volume of 50 ml and a surface area of 75 cm$^2$. Each consisted of a length of gas-permeable silicone rubber tubing (AP202/60 - 35 mm inner diameter by 1.5 mm wall, Advanced Polymers Ltd, UK) sealed at both ends with butyl rubber bungs. A suitable length of silicone tubing was passed through a hole in one of the bungs and capped with a stopcock to allow sampling at the surface. Chambers were encased in plastic mesh to protect the membrane during installation.



Typical of similar designs, soil gas equilibration chambers were capable of equilibrating with the external atmosphere in less than 24 h (Holter, 1990; Jacinthe and Dick, 1996; Kammann et al., 2001).

Soil volumetric water content was determined from triplicate measurements in the upper 6 cm of soil using a ML2x ThetaProbe (Delta-T Ltd., UK). Water-filled pore space (WFPS) was calculated from these data using estimates of porosity in the upper 10 cm based on plot averaged bulk density and site averaged particle density measurements (Table 1). Soil temperature was determined from triplicate measurements at 5 cm using a type K thermocouple penetration probe (Omega Engineering Ltd.,UK).

Available inorganic ammonium ($NH_4^+$) and nitrate ($NO_3^-$) concentrations were determined from resin bags buried in the the upper 10 cm of soil (Giblin et al., 1994; Templer et al., 2005). Each resin bag consisted of 5 g of ion exchange resin (Dowex Marathon MR-3, Sigma Aldrich) encased in a lycra bag. Resin bags were collected and replaced with a new bag during each plot visit. Following Templer et al. (2005), inorganic nitrogen was extracted from collected resin bags using 2 N KCL and it's concentration determined colorimetrically using a Burkard SFA2 (Burkard Scientific Ltd., Uxbridge, UK) continuous flow analyser at the University of Aberdeen. Reported available inorganic $NH_4^+$ and $NO_3^-$ concentrations are normalised to the amount of resin from which they were extracted and their deployment period, and reported as µg $NH_4^+$ - N g$^{-1}$ resin d$^{-1}$ and µg $NO_3^-$ - N g$^{-1}$ resin d$^{-1}$.

### 2.5 Statistical analyses

Statistical analysis was conducted in R version 3.1.1 (R Core Team, 2013). Linear mixed effect models were used to test the influence of forest type and season on measured variables as the dataset is unbalanced, with fewer measurements in the premontane forest, and nested within sampling month across forest types and in replicate plots within forest type (Pinheiro and Bates, 2000). In this respect, random intercept linear mixed effect models computed using the NLME package were used to test the effect of forest type and season on monthly plot means of net $CH_4$ flux, $CO_2$ flux, soil $O_2$ concentration, WFPS, soil temperature, available $NH_4^+$ and available $NO_3^-$ with forest type and season as fixed effects and sampling month and year as a random effect (Pinheiro et al., 2014). Following model fits, multiple comparison of site and season was conducted in the multcomp package with Tukey contrasts (Hothorn et al., 2008). Time-series of monthly plot means for these variables are provided in Fig. S1 to S7 of the supplementary material. Spatial and temporal relationships between measured variables were investigated using Pearson's correlation coefficient in the Hmisc package (Harrell et al., 2015). Spatial correlations were tested on dataset plot means whilst temporal correlations were applied to monthly site means calculated, in both cases, from monthly plot means. The validity of parametric tests was confirmed through visual inspection of residuals and as a result available $NH_4^+$ and available $NO_3^+$ were square root transformed in all reported statistical analyses to reduce heteroscedacity (Zuur et al., 2007). Statistical significance is reported at $p < 0.05$ unless stated otherwise.



## 3 Results

### 3.1 Variability in gas fluxes and soil environmental conditions

Fluxes of $CH_4$ were significantly influenced by forest type with larger fluxes at lower elevation (Table 2). All the forest types acted as a net sink for atmospheric $CH_4$ with mean (standard error) net $CH_4$ fluxes for dry and wet season of -1.6 (0.1) and

-1.1 (0.1) mg $CH_4$ - C $m^{-2}$ $d^{-1}$ in the upper montane forest, -1.1 (0.1) and -1.0 (0.1) mg $CH_4$ - C $m^{-2}$ $d^{-1}$ in the lower montane forest and -0.2 (0.1) and -0.1 (0.1) mg $CH_4$ - C $m^{-2}$ $d^{-1}$ in the premontane forest. During the dry season, net $CH_4$ fluxes varied significantly among all forest types. During the wet season, net $CH_4$ fluxes from premontane forest were significantly larger than those from both the upper and lower montane forests. Within forest types, no significant differences were identified between the dry and wet seasons when data from all time points were aggregated together. However, monthly time-series

from the upper montane forest indicate seasonal variability of exchange at this site, with a modest shift towards more negative $CH_4$ fluxes with the progression of the dry season (Fig. 2 a). Uptake dominated soil-atmosphere exchange in the upper and lower montane forests with emissions accounting for only 1 and 2 % of monthly mean $CH_4$ fluxes, respectively. In contrast, whilst net uptake was also evident, $CH_4$ emissions were more common in the premontane forest with 29 % of fluxes registering net $CH_4$ efflux.

Fluxes of $CO_2$ were significantly influenced by forest type with larger fluxes at lower elevation (Table 2). Fluxes of $CO_2$ for aggregated dry and wet season months were 2.9 (0.3) and 4.0 (0.3) g $CO_2$ - C $m^{-2}$ $d^{-1}$ in the upper montane forest, 4.3 (0.3) and 4.1 (0.3) g $CO_2$ - C $m^{-2}$ $d^{-1}$ in the lower montane forest and 5.2 (0.3) and 5.1 (0.3) g $CO_2$ - C $m^{-2}$ $d^{-1}$ in the premontane forest (Fig. 2 b). Dry season $CO_2$ fluxes were significantly smaller in the upper montane forest than both the premontane

forests, whilst, during the wet season no significant differences were identified. Within forest types, there were no significant seasonal differences in $CO_2$ flux but some evidence of wet season increases from the upper montane forest is apparent in the monthly time-series (Fig. 2 b).

Soil $O_2$ concentration at 10 cm soil depth was significantly influenced by forest type with slightly greater concentrations at

lower elevation (Table 2). Soil $O_2$ concentrations for aggregated dry and wet season months were 18.6 (0.3) and 18.9 (0.2) % in the upper montane forest, 19.1 (0.3) and 19.2 (0.2) % in the lower montane forest and 19.7 (0.3) and 19.8 (0.3) % in the premontane forest (Fig. 2 c). In both wet and dry season, soil $O_2$ concentration was significantly smaller in the upper montane than the premontane forest. However, these differences were marginal with a range of 1.1 %. Within forest types no significant differences in soil $O_2$ concentration between seasons were identified and little temporal variability is apparent in

the monthly time-series (Fig. 2 c).

WFPS was significantly influenced by forest type and its interaction with season, with greater saturation at lower elevation and during the wet season (Table 2). Mean WFPS for aggregated dry and wet season months was 24.4 (2.0) and 43.7 (1.7) %





in the upper montane forest, 35.4 (2.0) and 44.8 (1.7) % in the lower montane forest and 50.9 (2.1) and 53.3 (1.9) % in the premontane forest. Dry season WFPS was significantly different between all forest types; whilst during the wet season WFPS in the premontane forest was significantly greater than those from both the upper and lower montane forests. Within forest types, WFPS was significantly greater for wet than dry season for both the upper and lower montane forests as characterised by strong seasonality apparent in the monthly time-series for these sites (Fig. 2 d).

Soil temperature was significantly influenced by forest type and its interaction with season, with greater temperatures at lower elevation and during the wet season (Table 2). Mean soil temperature for aggregated dry and wet season months was 11.0 (0.2) and 11.9 (0.1) °C in the upper montane forest, 17.3 (0.2) and 18.0 (0.1) °C in the lower montane forest and 20.4 (0.2) and 20.7 (0.2) °C in the premontane forest (Fig. 2 e). In both seasons, soil temperatures were significantly different between all forest types. Within forest types, soil temperatures were significantly greater during the wet than dry season in both the upper and lower montane forests (Fig. 2 e).

Variation in the availability of inorganic nitrogen differed between species (Fig. 3). Available $NH_4^+$ was not significantly influenced by forest type or season (Table 2). Mean aggregated dry and wet season concentrations ranged from 7.8 to 10.8 µg $NH_4^+$ - N $g^{-1}$ resin $d^{-1}$ and 14.8 to 18.6 µg $NH_4^+$ - N $g^{-1}$ resin $d^{-1}$, respectively. In contrast, available $NO_3^-$ was significantly influenced by forest type with greater availability at lower elevations (Table 2). Mean available $NO_3^-$ for aggregated dry and wet season months was 0.4 (0.1) and 1.1 (0.0) µg $NO_3^-$ - N $g^{-1}$ resin $d^{-1}$ in the upper montane forest, 6.4 (0.1) and 9.6 (0.1) 21.0 (0.1) µg $NO_3^-$ - N $g^{-1}$ resin $d^{-1}$ in the lower montane forest and 21.2 (0.1) µg $NO_3^-$ - N $g^{-1}$ resin $d^{-1}$ in the premontane forest. Available $NO_3^-$ was significantly different between all forest types with no significant differences between season, in part due to considerable within plot variability, within forest type.

### 3.2 Spatial relationships between gas fluxes and environmental conditions

Across forest types, plot means of net $CH_4$ flux were significantly negatively correlated with elevation (Pearson's r = -0.81, p < 0.01, n = 9) and soil porosity (Pearson's r = -0.72, p = 0.03, n = 9 ) (Table 3). Similarly, the plot mean of net $CH_4$ flux was significantly positively correlated with the plot means of WFPS (Pearson's r = 0.84, p < 0.01, n = 9), soil temperature (Pearson's r = 0.83, p < 0.01, n = 9) and soil $O_2$ concentration (Pearson's r = 0.73, p = 0.03, n = 9), available $NH_4^+$ (Pearson's r = 0.71, p = 0.03, n = 9) and available $NO_3^-$ (Pearson's r = 0.88, p = < 0.01, n = 9), respectively. There was no significant relationship between the plot mean of net $CH_4$ flux and the plot mean of $CO_2$ flux (Pearson's r = - 0.61, p = 0.08, n = 9). Significant co-correlations exist between the plot means of many measured environmental variables. For example, strong correlations (i.e. p < 0.01) exist between plot means of net $CH_4$ flux and elevation, WFPS, soil temperature and available $NO_3^-$ (Fig. 4). However, both WFPS (Pearson's r = -0.79, p = 0.01, n = 9) and soil temperature (Pearson's r = -0.94, p < 0.01, n = 9) are significantly negatively correlated with elevation and are also significantly positively correlated with each other (Pearson's r = 0.94, p = 0.01, n < 9).





### 3.3 Temporal relationships between gas fluxes and environmental conditions

The drivers of temporal variability in net $CH_4$ flux varied within forest types. For example, in the upper montane forest, monthly site mean net $CH_4$ flux was significantly positively correlated with WFPS (Pearson's r = 0.54, p < 0.01, n = 28) and soil temperature (Pearson's r = 0.52 , p < 0.01, n = 27). Similarly to spatial co-correlations observed across forest types,

monthly site means of WFPS and soil temperature (Pearson's r = 0.60, p < 0.01, n = 27) were positively correlated with each other in the upper montane forest. In contrast, in the lower montane forest, monthly site mean net $CH_4$ flux was significantly negatively correlated with $CO_2$ flux (Pearson's r = -0.70, p < 0.01, n = 29). Whilst, in the premontane forest, no significant correlations between monthly site mean net $CH_4$ flux and other measured variables were identified (Table 4). For example, the strongest relationship with net $CH_4$ flux was a positive correlation with monthly site mean WFPS (Pearson's r = 0.33, p =

0.16, n = 19).

### 4 Discussion

### 4.1 Uptake of $CH_4$ by Andean forest soils in southern Peru

Upper montane, lower montane and premontane forests in the southern tropical Andes of Peru principally acted as sinks for atmospheric $CH_4$ (Fig. 2 a). Seasonal mean net $CH_4$ fluxes from these soils ranged from -1.6 to -0.1 mg $CH_4$ - C m$^{-2}$ d$^{-1}$,

indicating that soil-atmosphere $CH_4$ exchange in these forests are comparable to those previously reported for these sites and similar environments elsewhere. The major difference between the exchange rates reported here and the preliminary analysis of these data by Teh et al. (2014) is that the longer time-series indicates that these premontane forests act as a net sink rather than source of atmospheric $CH_4$ during the wet season. Reported mean net $CH_4$ fluxes for tropical forest soils above 600 m asl range from -1.6 to -0.2 mg $CH_4$ - C m$^{-2}$ d$^{-1}$ for the northern Andes in Ecuador (Wolf et al., 2012), -0.9 to -0.2 mg $CH_4$ - C

m$^{-2}$ d$^{-1}$ for central Sumatra and Sulawesi in Indonesia (Ishizuka et al., 2005b; Purbopuspito et al., 2006), -0.1 to 0.0 mg $CH_4$ - C m$^{-2}$ d$^{-1}$ for Mayombe highlands in the Republic of Congo (Delmas et al., 1992), -0.7 mg $CH_4$ - C m$^{-2}$ d$^{-1}$ for a tableland in northern Australia (Kiese et al., 2008), -1.4 mg $CH_4$ - C m$^{-2}$ d$^{-1}$ in Kenya (Werner et al., 2007), -0.5 mg $CH_4$ - C m$^{-2}$ d$^{-1}$ in China (Werner et al., 2006), -0.1 mg $CH_4$ - C m$^{-2}$ d$^{-1}$ in Panama (Veldkamp et al., 2013)  and -1.2 mg $CH_4$ - C m$^{-2}$ d$^{-1}$ for Atlantic forest in Brazil (Sousa Neto et al., 2011). Similarly, mean soil-atmosphere $CH_4$ exchange rates for lowland tropical

forests in South America have been reported in the range of -1.4 to -0.1 mg $CH_4$ - C m$^{-2}$ d$^{-1}$ (Davidson et al., 2004, 2008; Fernandes et al., 2002; Keller et al., 1986, 2005; Sousa Neto et al., 2011; Steudler et al., 1996; Verchot et al., 2000).

Within forest types, net $CH_4$ fluxes were not significantly different between wet and dry season when data were aggregated together by season, indicating that these forests show little overall seasonal variability in net $CH_4$ uptake (Table 2). The only

exception to this was the upper montane forest, where we see a strengthening of net $CH_4$ uptake with the procession of the dry season; equating to a ~ 30 % decrease in net $CH_4$ uptake between dry and wet season. Our inability to detect statistically





significant difference between seasons at this site may reflect interannual variability, and the fact environmental conditions change gradually across seasonal transitions (Clark et al., 2014). The aseasonality in $CH_4$ exchange in the lower montane and premontane forest conforms with observations from tropical montane forests in Ecuador and Indonesia with aseasonal climates, while the behaviour of the upper montane forests appears more similar to that of lowland tropical forests with

seasonal climates (Davidson et al., 2008; Keller et al., 2005; Purbopuspito et al., 2006; Verchot et al., 2000; Wolf et al., 2012). This contrast between the upper montane forest on one hand, and lower montane and premontane forest on the other, may reflect a threshold associated with edaphic conditions and decreases in total precipitation and temperature with elevation across our transect.

We did not observe notable spatial or temporal hotspots of $CH_4$ emission in these forests, contrasting with observations from regions such as the Caribbean and Hawaii (Schuur et al., 2001; Silver et al., 1999; Teh et al., 2005). For example, Silver et al. (1999) report mean net $CH_4$ fluxes of 0.2 to 73.2 mg $CH_4$ - C m$^{-2}$ d$^{-1}$ from montane ecosystems in Puerto Rico . Indeed, source activity in the upper and lower montane forests of this study represented only 1 - 2 % of fluxes. Emissions were more prevalent in the premontane forest, accounting for 29 % of fluxes, suggesting that emission hotspots are possible in these

soils but may not have been captured by our sampling strategy (Davidson et al., 2004; Delmas et al., 1992; Silver et al., 1999).

### 4.2 Environmental controls on net $CH_4$ fluxes in tropical Andean forests of southern Peru

The decrease in net $CH_4$ flux with elevation across the transect is strongly related to decreases in available $NO_3^-$, WFPS and soil temperature (Fig. 4). As there is considerable covariance between these environmental conditions a number of plausible,

but confounded, mechanisms may explain why $CH_4$ uptake is greatest in the upper montane forest and smallest in the premontane forest (Table 3). The predominance of $CH_4$ uptake and the lack of evidence for widespread anoxia, with soil $O_2$ measurements typically in excess of 15 %, indicates that net $CH_4$ exchange is dominated by the activity of high affinity methanotrophs which are unlikely to be sensitive to $O_2$ availability (Bender and Conrad, 1992; Teh et al., 2006). Indeed, unlike studies where landscapes exhibiting net source activity are considered we do not find evidence that decreases in $O_2$

concentration are driving net $CH_4$ exchange (Silver et al., 1999; Teh et al., 2014). The positive relationship between available $NO_3^+$ and net CH4 flux may suggest that $CH_4$ uptake is inhibited by $NO_3^-$. This observation supports previous, albeit poorly understood, observations that $CH_4$ oxidation is more sensitive to the presence of $NO_3^-$ rather than $NH_4^+$ (Mochizuki et al., 2012; Reay and Nedwell, 2004). Interestingly, this negative relationship between $CH_4$ uptake and $NO_3^-$ is the reverse of that observed across a similar transect in the Ecuadorian Andes (Wolf et al., 2012). The positive relationship between net $CH_4$

flux and WFPS across forest types conforms to the expectation that high-affinity methanotrophs are limited by $CH_4$ supply in response to diffusional constraints imposed by soil structure and water content (Curry, 2007; von Fischer et al., 2009; Smith et al., 2003). Notably, this relationship appears to be underpinned by dissimilarities in WFPS and net $CH_4$ flux across the upper and lower montane forest plots during the dry season. The importance of such spatial relationships between soil-





atmosphere $CH_4$ exchange and WFPS have previously been highlighted for lowland tropical soils and across the sites considered here but not in studies across other montane forests, where gravimetric water contents rather than WFPS have been reported (Purbopuspito et al., 2006; Teh et al., 2014; Verchot et al., 2000; Wolf et al., 2012). A positive relationship between net $CH_4$ flux and soil temperature was also identified. Given that we would expect metabolic rates to increase rather

than decrease in response to increased temperature, we suggest that this relationship results from covariance between soil temperature and available $NO_3^-$ or WFPS with elevation (Sousa Neto et al., 2011). Alternatively, if methanogenic activity, despite limited evidence for anoxia, is playing a sizeable role in determining net $CH_4$ flux across these forests this relationship could reflect the greater temperature sensitivity of methanogenesis relative to methanotrophy, leading to increased production at lower elevations (Segers, 1998). Similarly, in this situation, the positive relationship between WFPS

and net $CH_4$ flux could reflect the promotion of anoxic microsites, not captured by our $O_2$ measurements, by greater diffusional constraints (Verchot et al., 2000). However, counter to our observations we may also expect increases in available $NO_3^-$ to competitively suppress methanogenic activity (Chidthaisong and Conrad, 2000).

The controls on temporal variations within forest types in net $CH_4$ flux differed among forests. As with the comparison

across forest types, we find little support for soil $O_2$ concentration as a important factor in determining variability in net $CH_4$ fluxes. In upper montane forest, net $CH_4$ fluxes were best explained by a positive correlation with WFPS, indicating that wet season increases in soil moisture act to limit the diffusion $CH_4$ to methanotrophic communities (Table 4). Similarly to spatial relationships observed across these forests, a strong positive relationship between temporal variations in net $CH_4$ flux and soil temperature was also observed for this site, presumably because of covariance, with wetter, warmer conditions from

October to April, between WFPS and soil temperature. Interestingly, positive correlation between net $CO_2$ flux and WFPS at this site suggests that soil respiration, but not $CH_4$ uptake, may be water limited during the dry season. Given the importance of the contribution of root and litter respiration to total soil respiration in this forest this may reflect differences in the sensitivity of the communities involved; for example, microbial communities in the litter may experience greater water limitations than those in the soil (Zimmermann et al., 2009b, 2010b). In the lower montane forest, increases in net $CH_4$ flux

were related to decreases in $CO_2$ flux indicating that conditions favourable for methanotrophy may be similar to those for general soil respiration (Purbopuspito et al., 2006; Wolf et al., 2012). Significant differences in WFPS between wet and dry season were not reflected in net $CH_4$ fluxes from these soils suggesting these variations were not great enough to sufficiently limit diffusion of $CH_4$ as to constrain uptake rates. In the premontane forest no drivers of net $CH_4$ flux were identifiable reflecting fewer observations from this site and the apparent lack of seasonality in any of the measured parameters

(Purbopuspito et al., 2006; Sousa Neto et al., 2011; Wolf et al., 2012).

### 4.3 Altitudinal trends in soil $CH_4$ cycling across tropical montane forests

With the exception of strong source activity in some island settings, rates of $CH_4$ uptake by tropical montane forest soils are broadly comparable across tropical montane and lowland locations globally. However, within montane regions, the





relationship between soil-atmosphere $CH_4$ exchange and elevation is poorly constrained (Table 5). For example, differing patterns in soil-atmosphere $CH_4$ exchange are reported here compared to that of altitudinal transects in Ecuador (Wolf et al., 2012) or Indonesia (Purbopuspito et al., 2006). The relationship between elevation and edaphic conditions is broadly similar for these studies with upper montane, lower montane and premontane forests occurring within similar elevation bands (Foster, 2001). In this respect, montane and premontane forest soils in these studies are differentiated by the presence of thick organic horizons at the surface. Despite this similarity, differing relationships between net $CH_4$ flux and elevation are apparent for the soils of Peru and Ecuador, with the former showing greatest uptake in the upper montane soils, while the latter shows greatest uptake in premontane soils (Wolf et al., 2012). Furthermore, uptake peaked in the lower montane soils in Indonesia (Purbopuspito et al., 2006). From these contrasts, it is possible to suggest that relationships with temperature identified here and in Ecuador, as previously discussed, result from covariance with some other driver rather than as a result of the temperature sensitivity of $CH_4$ uptake (Wolf et al., 2012). Similarly, differing relationships between net $CH_4$ fluxes and the availability of inorganic nitrogen do not allow for simple generalisations to be drawn across these ecosystems. For example, in this study, a positive relationship between net $CH_4$ flux and available $NO_3^-$ suggests inhibition of uptake; while in Ecuador, a negative relationship between these parameters led Wolf (2012) to hypothesize that uptake is nitrogen limited. In contrast to both of these Andean studies, no relationship was identified in Indonesia (Purbopuspito et al., 2006). We speculate that the inverse altitudinal patterns in soil-atmosphere $CH_4$ exchange in Peru and Ecuador may reflect differences in their precipitation regimes and the influence of soil structure and water content on diffusion. Such a mechanism is supported by the positive relationship between net $CH_4$ flux and WFPS in this study. The fact that no relationship between net $CH_4$ flux and water content were identified in either Ecuador or Indonesia may reflect the fact that both of these studies measured gravimetric water content rather than water-filled pore space, where the former moisture index does not always adequately characterise the influence of soil structure on diffusion (Purbopuspito et al., 2006; Verchot et al., 2000; Wolf et al., 2012) . Wolf et al. (2012) highlight that organic horizons of these montane forests are active zones of methanotrophy. It seems likely that a better understanding of the edaphic controls, relating soil structure and precipitation, on methanotrophy in such soils is required to reconcile differences in soil atmosphere $CH_4$ exchange across such landscapes.

## 5 Conclusion

The findings of this study suggest that the upper montane, lower montane and premontane forests of south-eastern Peru principally act as sinks for atmospheric $CH_4$. Uptake rates in these soils are comparable to activity observed globally for both montane and lowland tropical forests. Uptake rates were greatest in the upper montane forest and lowest in the premontane forest. We find that across the landscape, these soils are predominantly oxic and soil $CH_4$ cycling is likely dominated by the activity of high affinity methanotrophs. In this regard, strong positive relationships between net $CH_4$ flux and both available $NO_3^-$ and WFPS were identified suggesting that variations in $CH_4$ uptake across the landscape may be driven by $NO_3^-$ inhibition and/or constraints on the diffusional ingress of $CH_4$ from the atmosphere. Despite distinct wet and dry seasons in



this region, evidence for seasonality in net $CH_4$ fluxes were only identified in the upper montane forest soils. The increase in $CH_4$ uptake with elevation differs with that previously reported for similar environments in Ecuador and Indonesia, suggesting that an improved understanding of the controls on methanotrophy in the organic horizons of tropical montane forest soils are required.

**Acknowledgements**

This study is a product of the Andes Biodiversity and Ecosystem Research Group consortium (http://www.andesconservation.org/). The authors would like to acknowledge the agencies that funded this research; the UK Natural Environment Research Council (NERC; joint grant references NE/G018278/1, NE/H006583, NE/H007849 and NE/H006753) and the Norwegian Agency for Development Cooperation (Norad; via a sub-contract to Yit Arn Teh managed

by the Amazon Conservation Association). Patrick Meir was also supported by an Australian Research Council Fellowship (FT110100457). Javier Eduardo Silva Espejo, Walter Huaraca Huasco and the ABIDA NGO provided critical fieldwork and logistical support. Angus Calder, Michael Mcgibbon, Vicky Munro and Nick Morley provided invaluable laboratory support. Thanks to Adrian Tejedor and the Amazon Conservation Association (http://www.amazonconservation.org/), who provided assistance with site access and site selection at Hacienda Villa Carmen. This publication is a contribution from the Scottish

Alliance for Geoscience, Environment and Society (http://www.sages.ac.uk).

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



**Table 1** Study site and plot characteristics.

| Study sites | | | | | | Study plots | | | | |
|---|---|---|---|---|---|---|---|---|---|---|
| Field station | Forest type | Longitude | Latitude | MAT[a] | MAP[a] | Elevation | Topography | Bulk density[b] | Particle density[b] | Porosity[b] |
| | | (°S) | (°W) | (°C) | (mm) | (m asl) | | (g cm$^{-2}$) | (g cm$^{-2}$) | |
| Villa Carmen | Pre-montane | 12°53'43" | 71°24'04" | 23.4 | 5300 | 1070 to 1088 | ridge, slope & flat | 0.27 to 0.31 | 2.2 | 0.86 to 0.88 |
| San Pedro | Lower montane | 13°02'56" | 71°32'13" | 18.8 | 2600 | 1532 to 1768 | ridge, slope & flat | 0.09 to 0.22 | 1.7 | 0.87 to 0.95 |
| Wayqecha | Upper montane | 13°11'24" | 71°35'13" | 12.5 | 1700 | 2811 to 2962 | ridge & slopes | 0.08 to 0.11 | 1.4 | 0.92 to 0.94 |

[a]mean annual air temperature and mean annual precipitation & [b]soil properties in the upper 10 cm.



**Table 2** Forest type means and standard errors for aggregated dry (May - September) and wet (October - April) season months. Capital letters indicate significant differences (p < 0.05) among forest types within season and lower case letters indicate significant differences between season within forest types.

| Forest type | Net $CH_4$ flux (mg C m$^{-2}$ d$^{-1}$) | | Net $CO_2$ flux (g C m$^{-2}$ d$^{-1}$) | | $O_2$ concentration (%) | | WFPS (%) | | Soil temperature (°C) | | Available $NH_4^+$ (µg N g$^{-1}$ res. d$^{-1}$) | | Available $NO_3^-$ (µg N g$^{-1}$ res. d$^{-1}$) | |
|---|---|---|---|---|---|---|---|---|---|---|---|---|---|---|
| | Dry | Wet | Dry | Wet | Dry | Wet | Dry | Wet | Dry | Wet | Dry | Wet | Dry | Wet |
| Pre-montane | -0.2 (0.1)$^A$ | -0.1 (0.1)$^A$ | 5.2 (0.3)$^A$ | 5.1 (0.3)$^A$ | 19.7 (0.3)$^A$ | 19.8 (0.3)$^A$ | 50.9 (2.1)$^A$ | 53.3 (1.9)$^A$ | 20.4 (0.2)$^A$ | 20.7 (0.2)$^A$ | 10.0 (0.6) | 17.9 (0.05) | 21.0 (0.1)$^A$ | 21.2 (0.1)$^A$ |
| Lower montane | -1.1 (0.1)$^B$ | -1.0 (0.1)$^B$ | 4.3 (0.3)$^{AB}$ | 4.1 (0.3)$^A$ | 19.1 (0.3)$^{AB}$ | 19.2 (0.2)$^{AB}$ | 35.4 (2.0)$^{Ba}$ | 44.8 (1.7)$^{Bb}$ | 17.3 (0.2)$^{Ba}$ | 18.0 (0.1)$^{Bb}$ | 7.8 (0.6) | 14.8 (0.5) | 6.4 (0.1)$^B$ | 9.6 (0.1)$^B$ |
| Upper montane | -1.6 (0.1)$^C$ | -1.1 (0.1)$^B$ | 2.9 (0.3)$^B$ | 4.0 (0.3)$^A$ | 18.6 (0.3)$^B$ | 18.9 (0.2)$^B$ | 24.4 (2.0)$^{Ca}$ | 43.7 (1.7)$^{Bb}$ | 11.0 (0.2)$^{Ca}$ | 11.9 (0.1)$^{Cb}$ | 10.8 (0.6) | 18.6 (0.5) | 0.4 (0.1)$^C$ | 1.1 (0.0)$^C$ |



**Table 3** Pearson's correlation coefficient matrix for dataset plot means of measured variables among forest types. ** $p <$ 0.01, * $p < 0.05$.

| Variable pairs | Net $CH_4$ flux | Net $CO_2$ flux | $O_2$ concentration | WFPS | Soil temperature | √Available $NH_4^+$ | √Available $NO_3^-$ |
|---|---|---|---|---|---|---|---|
| Net $CO_2$ flux | 0.61 | - | - | - | - | - | - |
| $O_2$ concentration | 0.73* | 0.55 | - | - | - | - | - |
| WFPS | 0.84** | 0.43 | 0.51 | - | - | - | - |
| Soil temperature | 0.83** | 0.66 | 0.77* | 0.79* | - | - | - |
| √Available $NH_4^+$ | 0.71* | 0.36 | 0.31 | 0.41 | 0.25 | - | - |
| √Available $NO_3^-$ | 0.88** | 0.67* | 0.72* | 0.83** | -0.94** | 0.42 | - |
| Elevation | -0.81** | -0.66 | -0.76* | -0.79* | -1.00** | -0.22 | -0.94** |
| Porosity | -0.72* | -0.75* | -0.59 | -0.68* | -0.80* | -0.23 | -0.86** |





**Table 4** Pearson's correlation coefficient matrix for monthly site means of measured variables within forest types. ** $p < 0.01$, * $p < 0.05$.

| Forest type | Variable pairs | Net $CH_4$ flux | Net $CO_2$ flux | $O_2$ concentration | WFPS | Soil temperature | $\sqrt{}$Available $NH_4^+$ |
|---|---|---|---|---|---|---|---|
| Premontane | Net $CO_2$ flux | -0.21 | - | - | - | - | - |
| | $O_2$ concentration | 0.05 | 0.33 | - | - | - | - |
| | WFPS | 0.33 | -0.34 | -0.44 | - | - | - |
| | Soil temperature | -0.16 | 0.13 | -0.44 | -0.49* | - | - |
| | $\sqrt{}$Available $NH_4^+$ | -0.05 | 0.00 | 0.47 | -0.38 | 0.11 | - |
| | $\sqrt{}$Available $NO_3^-$ | -0.25 | 0.20 | 0.46 | -0.46 | 0.03 | 0.93** |
| Lower montane | Net $CO_2$ flux | -0.70** | - | - | - | - | - |
| | $O_2$ concentration | -0.20 | 0.08 | - | - | - | - |
| | WFPS | 0.16 | 0.15 | -0.23 | - | - | - |
| | Soil temperature | 0.08 | -0.08 | -0.29 | 0.12 | - | - |
| | $\sqrt{}$Available $NH_4^+$ | 0.04 | -0.20 | 0.37 | -0.11 | 0.40 | - |
| | $\sqrt{}$Available $NO_3^-$ | -0.14 | 0.06 | 0.53 | 0.22 | 0.35 | 0.24 |
| Upper montane | Net $CO_2$ flux | -0.20 | - | - | - | - | - |
| | $O_2$ concentration | 0.04 | 0.09 | - | - | - | - |
| | WFPS | 0.54** | 0.49** | 0.09 | - | - | - |
| | Soil temperature | 0.52** | 0.29 | -0.21 | 0.60** | - | - |
| | $\sqrt{}$Available $NH_4^+$ | 0.19 | 0.11 | 0.03 | -0.10 | 0.11 | - |
| | $\sqrt{}$Available $NO_3^-$ | -0.02 | 0.46 | -0.32 | 0.53* | 0.54* | 0.39 |



**Table 5** Characteristics and annual mean (standard error) net $CH_4$ fluxes reported for montane forests in Peru (this study), Ecuador (Wolf et al., 2012) and Indonesia (Purbopuspito et al., 2006).

| Country | Forest type | Elevation (m asl) | Organic horizon thickness (cm) | MAT[a] (°C) | MAP[a] (mm) | Net $CH_4$ flux (mg $CH_4$ - C $m^{-2}$ $d^{-1}$) |
|---|---|---|---|---|---|---|
| Ecuador | Premontane | 900 to 1200 | 2.5 to 6.5 | 19.4 | 2230 | -1.5 |
| | Lower montane | 1800 to 2100 | 4.0 to 24.0 | 15.7 | 1950 | -0.9 |
| | Upper montane | 2800 to 3000 | 6.6 to 22.2 | 9.4 | 4500 | -0.3 |
| Peru | Premontane | 1070 to 1088 | < 5 | 23.4 | 5300 | -0.2 |
| | Lower montane | 1532 to 1768 | ~ 10 | 18.8 | 2600 | -1.1 |
| | Upper montane | 2811 to 2962 | ~ 20 | 12.5 | 1700 | -1.3 |
| Indonesia | Premontane | 1190 | 0 | 22.5 | 1500 | -0.7 |
| | Lower montane | 1800 | 15 to 25 | 18.3 | - | -0.9 |
| | Upper montane | 2470 | 10 to 20 | 14.6 | - | -0.4 |

[a]mean annual air temperature and mean annual precipitation



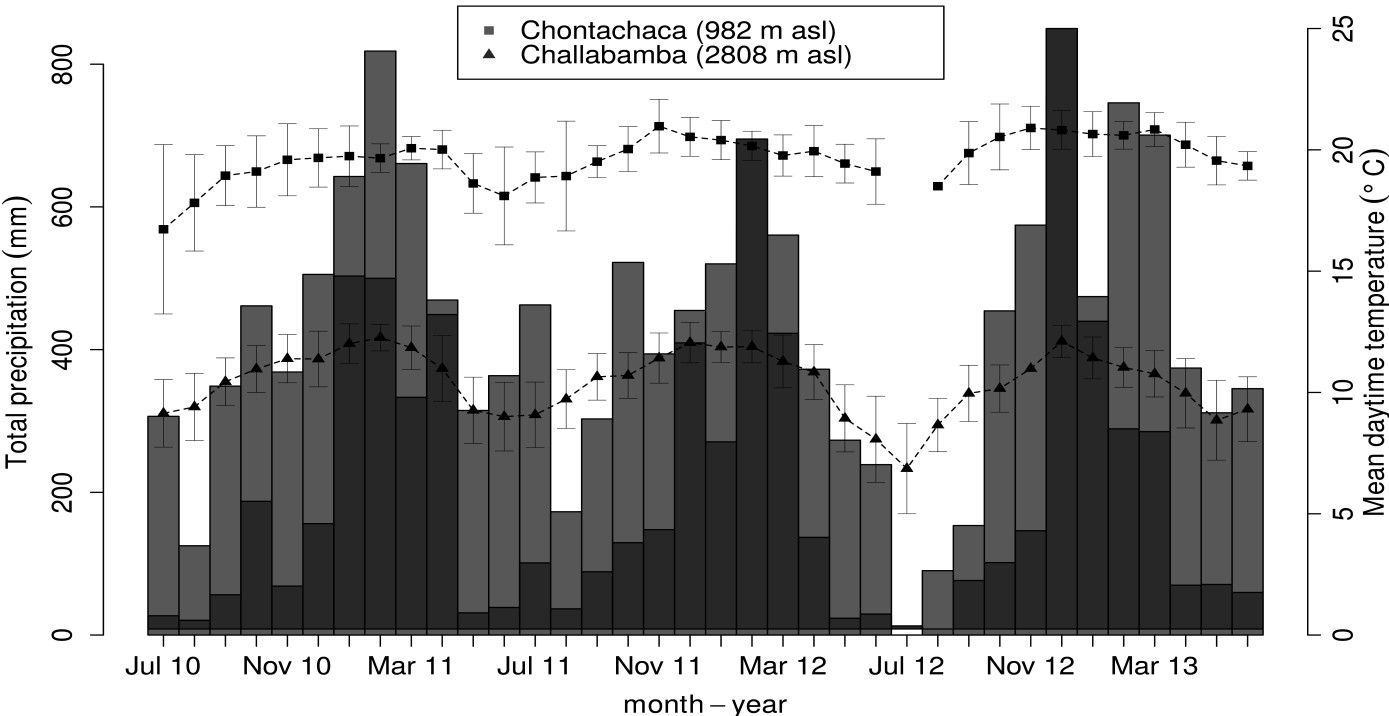

**Figure 1** Total monthly precipitation and monthly mean day time temperature between July 2010 and June 2013 at 982 m asl (Chontachaca weather station: 13°01′26″S 71°28′04″W) and 2808 m asl (Challabamba weather station: 13°13′03″S 71°38′50″W). Temperature error bars are standard errors. No data was available for July 2012 at Contachaca. Plotted data was retrieved from the Servicio Nacional De Meteorgia e Hidrologia Del Peru (http://www.senamhi.gob.pe).





**Figure 2** Monthly site means and standard deviation of a) net $CH_4$ flux, b) net $CO_2$ flux, c) soil $O_2$ concentration, d) WFPS and e) soil temperature. Shading indicates the wet season of October – April.




**Figure 3** Boxplots of monthly mean inorganic nitrogen availability a) available $NH_4^+$ and b) available $NO_3^-$.



**Figure 4** Relationships between dataset plot means of net CH$_4$ flux and a) elevation, b) square-root transformed available NO$_3^-$, c) soil temperature and d) WFPS across forest types. Error bars indicate standard error. Dashed lines indicate linear regressions between plotted variables: a) Net CH$_4$ flux = 0.49 × 10$^{-3}$ * elevation – 0.05  (r$^2$ = 0.61), b) Net CH$_4$ flux = 0.25 * √available NO$_3^-$ – 1.59 (r$^2$ = 0.74), c) Net CH$_4$ flux = 0.10 * soil temperature – 2.58 (r$^2$ = 0.64) and d)  Net CH$_4$ flux = 0.05 * WFPS – 3.07 (r$^2$ = 0.67).