# Peer review of "Drivers of atmospheric methane uptake by montane forest soils in the southern Peruvian Andes"

_Biogeosciences, 2016_

## Referee Comment (RC1) · Anonymous Referee #1 · 24 Feb 2016

General Comments Jones and co-authors present a very comprehensive and well-written study about rates of CH4 uptake by forest soils in the southern Peruvian Andes. Since there are only few ecosystem CH4 flux measurements from tropical montane forest regions, this study is very valuable and provides additional information on this subject. Interestingly, their results differ from other studies. Hence, there does not seem to be any general trend in CH4 fluxes along elevation gradients in tropical montane forest regions. Contrasting results make it very complicated but very fascinating. The authors try to explain this discrepancy but remain very speculative throughout the discussion. Wolf et al (2011) got a deeper insight into the soil "black box" by conducting incubations of soil samples from different soil horizons. Finally, they identified a stratification of CH4-uptake activity within the soil profile that highlights the heterogeneity of methane cycling processes in organic soils of tropical montane forests. After such

a study by Wolf et al. (2011), it would be nice to identify hot spots of CH4 consumption and/or production within soils of their region and how that correlates with available nitrate, ammonium, oxygen etc. Furthermore, the discussion about N-inhibition or N-limitation of CH4 consumption and/or production is very speculative without having any information about present methanotrophic or methanogenic community composition and/or activity, especially when the results are so different. Other processes, as well, may eventually lead to the observed positive correlation between net CH4 flux and nitrate concentrations. Dependent on the nutrient status of the respective forest type, increased soil nitrate availability may stimulate plant growth that accelerates organic carbon availability via root exudation for methanogens and other microorganisms and finally lead to an increase of CH4 production in anoxic microsites and a decrease of net CH4 consumption (see Bodelier et al. 2011). What is with phosphorus (see Wolf et al. 2011)? I think that nutrient status of the diverse vegetation including the deep roots within organic-rich soils of tropical montane forests may play an important role in structuring microbial community composition and activity that may be as important as soil structure and precipitation. However, the present study is very valuable and provides more information than countless artificial laboratory incubation studies but complementary incubations and a combined approach including microbiological and biogeochemical methods may have the potential to explain the underlying processes (see. Christiansen et al. (2015); McCalley et al. (2014)).

Specific Comments I would remove the word significantly throughout the text. It is in almost every sentence of the "Results" section. I think it is enough if you say that A is higher than B or A influences B. If something is not significant there is no difference or influence. Additionally, you define statistical significance at $p < 0.05$. That is enough, I think.

Page 7, Line 5+6: How did you measure particle density and porosity?

Page 4+5: Could you clarify how many plots were installed, in total?

Page 9, Line 22. . . and Table 3; Figure 4: As far as I understand, you have 3 plots per elevation (these are your independent samples if you say they were randomly selected; n=3). Now, you can do linear regression between your variables of interest among these three points but in my opinion you are not allowed to do linear regression among all samples (9 plot means) of the elevation gradient because they are not independent! You can check whether your forest type means differ from each other but not a linear regression among 9 plot means.

References Bodelier. 2011. Interactions between nitrogenous fertilizers and methane cycling in wetland and upland soil. Current Opinion in Environmental Sustainability 3: 379-388.

Christiansen et al. 2014. DOI: 10.1007/s10533-014-0026-7

McCalley et al. 2015. Methane dynamics regulated by microbial community response to permafrost thaw. doi: 10.1038/nature13798.

―――――――――――――――――

---

## Referee Comment (RC2) · Anonymous Referee #2 · 29 Feb 2016

General comments: This was a well-written study summarizing up to 2.5 years of soil-$CH_4$-flux monitoring along an elevation gradient of tropical montane forests in Peru. Supporting measurements included soil $CO_2$ fluxes, soil $O_2$, available N, moisture and temperature. The introduction provided a solid basis for the study, methods were described in detail and results were presented concisely. The figures and tables were well designed and informative. However, I would like to see the authors spend a little more time polishing the discussion; there were several instances where statements made were too vague to contribute anything substantial to what was being discussed (see below for specific cases). Altogether, the very interesting contrasting results of this study with those of previous studies makes this an important contribution to research focused on understanding soil $CH_4$ fluxes in tropical montane regions.

Specific comments: - Pg 11, lines 6-8: this sentence is rather vague - can you rephrase

it to state specifically what you think is occurring (perhaps with a reference)? - Pg 11, line 15: can you suggest what you might have done differently? - Pg 11, line 32: 'dissimilarities' is vague - please rephrase this sentence to explain specifically what you mean. - Pg 12, line 11-12: the purpose of this sentence is unclear, please rephrase and/or expand on what you are trying to say here. - Pg 13, line 10: discussed where? Are you referring to Pg. 11? If so, this is again quite vague. Can you suggest another possible driver? - Pg 13, line 17: what mechanism? You comment on differences and go on to suggest they may be related to soil structure, but can you specifically describe a possible mechanism? - Figure 4: Double-check your statistical theory here, but as they are currently presented, I don't think these graphs should have regression lines. I believe that once you choose to treat the 3 elevations as forest replicates, the within-elevation replicates would need to be averaged in order to avoid pseudoreplication.

Technical corrections: - Throughout the paper (most mistakes occur in the discussion) use only past tense rather than mixing past and present. - When citing references, it is more helpful for the reader if you put references directly behind the information they refer to instead of grouping them at the end of the sentence (i.e. Pg 11, line 5, Pg 12, line 3, Pg 13, line 21). - To avoid unnecessary confusion, consider using 'efflux' or 'emissions' to refer specifically to positive fluxes. For example, on Pg. 8, line 3, one could argue that larger fluxes actually occurred at the higher elevations. - Pg 8, line 3,16: at lower 'elevations' - Pg 8, line 11: use consistent language. Rather than saying 'negative fluxes', use 'uptake' as you do elsewhere. - Pg 9, line 2, Pg 10, line 31, Pg 13, line 13: A semi-colon is used to join two related (complete) sentences. Whilst or while are conjunctions, which can be used to compare two (normally contrasting) ideas in a single sentence. - Pg 9, line 14: by 'species' do you mean 'forest type'? - Pg 9, line 28: plot 'means' - Pg 10, line 2,8: delete 'for example' - Pg 10, line 28: delete 'together' - Pg 10, line 30: did you do this statistical comparison over time or only comparing wet vs dry season? If the latter is true, then you can't say "with the procession of. . ." - Pg 11, line 1: 'differences', the fact 'that' - Pg 11, lines 12-13, Pg 12, line 32: 'source activity' is awkward. Change to 'emissions' or 'efflux'. - Pg 11, line 32: 'underpinned' sounds

awkward here, perhaps 'supported'? - Pg 11, line 29-33 (and elsewhere): This part of the discussion would be improved if you indicated which tables/figures correspond to the results you are discussing. - Pg 12, line 17: 'Similar' to - Pg 12, lines 20, 26: 'the' positive correlations, 'the' wet and dry seasons - Pg 13, line 1: 'constrained' seems odd in this context, do you mean 'understood'? - Pg 13, line 4: move the reference to Table 5 from line 1 to here, after 'for these studies' - Pg 13, line 5: 'in this respect' doesn't fit here, can you reword?

---

## Author Comment (AC1) · 10 Apr 2016

Reponse to Reviewer 1

We would like to thank reviewer 1 for taking the time to provide a useful critique of the manuscript. We have responded to their concerns in blue and copied their original comments in black for ease of reference.

Wolf et al. (2011) got a deeper insight into the soil "black box" by conducting incubations of soil samples from different soil horizons. Finally, they identified a stratification of CH4-uptake activity within the soil profile that highlights the heterogeneity of methane cycling processes in organic soils of tropical montane forests. After such a study by Wolf et al. (2011), it would be nice to identify hot spots of CH4 consumption and/or production within soils of their region and how that correlates with available nitrate, ammonium, oxygen etc

We whole-heartedly agree with reviewer 1 that overcoming the inherent difficulties posed to sampling strategies in attempting to study hot spot activity remains a particularly interesting facet of understanding fine scale heterogeneity within tropical soils (see Hall et al. (2013) for a nice example).

Furthermore, the discussion about N-inhibition or N-limitation of CH4 consumption and/or production is very speculative without having any information about present methanotrophic or methanogenic community composition and/or activity, especially when the results are so different. Other processes, as well, may eventually lead to the observed positive correlation between net CH4 flux and nitrate concentrations. Dependent on the nutrient status of the respective forest type, increased soil nitrate availability may stimulate plant growth that accelerates organic carbon availability via root exudation for methanogens and other microorganisms and finally lead to an increase of CH4 production in anoxic microsites and a decrease of net CH4 consumption (see Bodelier et al. 2011)

We remain speculative as conceptual models linking $CH_4$ exchange and N availability are complicated and we are limited to inferring possible causes in terms of changes in net exchange and field conditions. The apparent differences between Indonesia, Ecuador and Peru discussed do indeed seem to suggest a better understanding of patterns in the underlying gross processes is required. We attempt to investigate, albeit at a crude scale, the potential for microsite methanogenic activity in influencing net exchange through measurement of bulk soil $O_2$ and net $CO_2$ fluxes. We have extended the text on Page 12 to accommodate the reviewer's good suggestion: *"However, we may also have expected increases in available $NO_3^-$ to competitively suppress methanogenic activity (Chidthaisong and Conrad, 2000). This is counter to the observation that net $CH_4$ is positively correlated to available $NO_3^-$ and that emissions are most prevalent in the premontane forest. Greater below-ground productivity at lower elevations (Girardin et al., 2010), potentially driven by greater nutrient availability and temperature, may also stimulate $CH_4$ production in the rhizosphere through the supply of labile substrates to methanogenic communities or maintenance of anaerobic microsites through the $O_2$ demand of heterotrophic respiration (Bodelier, 2011). Such a mechanism, not observed in this data, might be supported by a positive relationship between net $CH_4$ and $CO_2$ fluxes (Verchot et al., 2000)."*

What is with phosphorus (see Wolf et al. 2011)? I think that nutrient status of the diverse vegetation including the deep roots within organic-rich soils of tropical montane forests may play an important role in structuring microbial community composition and activity that may be as important as soil structure and precipitation.

In addition to available nitrate and ammonium, phosphate and nitrite data were also obtained from the resin bags. No significant relationship was found between nitrite and net $CH_4$ flux, whilst, a significant negative relationship was identified between phosphate. However, this relationship was less robust than those which form the main focus of our discussion (i.e. soil temperature, available nitrate and water-filled pore space). As understanding the influence of variations in microbial functional diversity is beyond the scope of our work we omitted these data for the sake of clarity. They will however be available when the dataset is archived with CEDA.

I would remove the word significantly throughout the text. It is in almost every sentence of the "Results" section. I think it is enough if you say that A is higher than B or A influences B. If something is not significant there is no difference or influence. Additionally, you define statistical significance at p<0.05. That is enough, I think.

Following this advice, we have altered the text of results section accordingly.

Page 7, Line 5+6: How did you measure particle density and porosity

We have added this information to Page 7: *"Plot bulk densities were determined from the weight of volumetric soil samples after oven drying at 105 °C for 24 hours. Forest type particle density was determined from measurement of bulked plot samples using a 10 ml pyncometer (Klute and others, 1986)."*

Page 4+5: Could you clarify how many plots were installed, in total

A total of 9 plots were installed; three plots were installed in each of the three forest types. See Page 5, Line 8 – 13:

*"Within each forest type three plots of 20 by 20 m were established approximately three months prior to the start of reported measurements in an attempt to minimise the effects of disturbances involved with installing sampling equipment (Varner et al., 2003). Within forest types the distance between plots ranged from ~ 100 to 1000 m. The plots in the premontane forest were each situated on a ridge, slope and flat feature between elevations of 1070 to 1088 m asl. Similarly, the lower montane forest plots were established on ridge, slope and flat features between elevations of 1532 to 1768 m asl. In the upper montane forest two plots were situated on slopes and the third on a ridge at elevations between 2811 to 2962 m asl."*

Page 9, Line 22 and Table 3; Figure 4: As far as I understand, you have 3 plots per elevation (these are your independent samples if you say they were randomly selected; n=3). Now, you can do linear regression between your variables of interest among these three points but in my opinion you are not allowed to do linear regression among all samples (9 plot means) of the elevation gradient because they are not independent! You can check whether your forest type means differ from each other but not a linear regression among 9 plot means.

Reviewer 2 also raised this point and we have copied this response there. Our decision to treat measurement plots within a 'forest type' (or elevation band)  as independent replicates of net $CH_4$ exchange is based on the assumption that spatial autocorrelation is limited to short distances (i.e. operating at sub- plot scales of ~ 1 to 10s of m). The plots in our study were more than 100 m apart. We treat our observations as longitudinal data to investigate the possible drivers of the relationship between net $CH_4$ flux and elevation within our study area. In an attempt to synthesise this information, we then discuss $CH_4$ exchange in terms of the ecosystem transitions (or 'forest types') seen across the landscape. This approach is adopted from the literature, for example, across montane forest landscapes (Purbopuspito et al., 2006, p.3) and more recently across lowland tropical forest landscapes (Hassler et al., 2015). We state our approach in the manuscript on Page 6, Line 19 – 21: *"Despite the three plots within each forest type broadly occurring within the same forest stand they were considered independent replicates of forest type as spatial correlations between net $CH_4$ fluxes in tropical forests are small (Ishizuka et al., 2005a; Purbopuspito et al., 2006)".* However, we acknowledge the concerns of both reviewers. The use of site (n = 3) or plot (n = 9) means in correlation tests does not fundamentally change the pattern or effect size of the relationships which form the basis of our discussion in section 4.2. For example, focussing on the most robust relationships identified between $CH_4$  exchange and environmental conditions (Table 3 (Pearson's r  > 0.8 , p < 0.05, n = 9) and then graphed in Figure 4):  n = 3, net $CH_4$ flux vs. elevation (Pearson's r = -0.85, p = 0.35), soil temperature (Pearson's r  = 0.86, p= 0.34), WFPS (Pearson's r = 0.99, p = 0.10) and NO3 (Pearson's r  = 0.92, p = 0.25). In an attempt to minimise confusion caused by the text, we have altered the somewhat unclear use of  'site' on Page 4 to fall into line with the way we treat the data and how the experimental approach along this transect has previously been described (e.g. Teh et al., 2014, p.2)

References
Hall, S. J., McDowell, W. H. and Silver, W. L.: When wet gets wetter: decoupling of moisture, redox biogeochemistry, and greenhouse gas fluxes in a humid tropical forest soil, Ecosystems, 16(4), 576–589, 2013.

Hassler, E., Corre, M. D., Tjoa, A., Damris, M., Utami, S. R. and Veldkamp, E.: Soil fertility controls soil–atmosphere carbon dioxide and methane fluxes in a tropical landscape converted from lowland forest to rubber and oil palm plantations, Biogeosciences, 12(19), 5831–5852, doi:10.5194/bg-12-5831-2015, 2015.

Purbopuspito, J., Veldkamp, E., Brumme, R. and Murdiyarso, D.: Trace gas fluxes and nitrogen cycling along an elevation sequence of tropical montane forests in Central Sulawesi, Indonesia, Global Biogeochemical Cycles, 20(3), doi:10.1029/2005GB002516, 2006.

Teh, Y. A., Diem, T., Jones, S., Quispe, L. H., Baggs, E., Morley, N., Richards, M., Smith, P. and Meir, P.: Methane and nitrous oxide fluxes across an elevation gradient in the tropical Peruvian Andes, Biogeosciences, 11, 2325–2339, 2014.

---

## Author Comment (AC2) · 10 Apr 2016

Reponse to Reviewer 2

We would like to thank Reviewer 2 for taking the time to provide a useful critique of the manuscript and in particular for providing detailed suggestions for improving the clarity of the discussion. We have responded to their concerns in blue and copied their original comments in black for ease of reference.

Pg 11, lines 6-8: this sentence is rather vague - can you rephrase it to state specifically what you think is occurring (perhaps with a reference)?

We have simplified the statement on Page 11: *"This may impart reflect the fact that seasonal variations in rainfall and temperature become more pronounced with elevation across this transect (Fig. 1)."* The issue is more appropriately discussed with reference to temporal relationship between net flux and WFPS in the following section. Page 12, Line ~16

Pg 11, line 15: can you suggest what you might have done differently?

We have expanded the text on Page 11: *"Emissions were more prevalent in the premontane forest, accounting for 29 % of fluxes, suggesting that emission hotspots are possible in these soils but may not have been captured by our sampling strategy (Delmas et al., 1992; Silver et al., 1999). Assessing and studying the potential for emissions from these ecosystems is likely to require higher resolution observations to capture spatial and temporal variability (Liptzin et al., 2011; Silver et al., 1999) combined with experimental manipulations (Hall et al., 2013) and a better understanding of below-ground $CH_4$ cycling (von Fischer and Hedin, 2007; Teh et al., 2005)."*

Pg 11, line 32: 'dissimilarities' is vague - please rephrase this sentence to explain specifically what you mean.

We have reworded the text on Page 11: *"Notably, this relationship appears to be driven by decreased WFPS and increased $CH_4$ uptake in the upper montane forest plot during the dry season."*

Pg 12, line 11-12: the purpose of this sentence is unclear, please rephrase and/or expand on what you are trying to say here.

We have expanded on text on Page 11: *"However, we may also have expected increases in available $NO_3^-$ to competitively suppress methanogenic activity (Chidthaisong and Conrad, 2000). This is counter to the observation that net $CH_4$ is positively correlated to available $NO_3^-$ and that emissions are most prevalent in the premontane forest. Greater below-ground productivity at lower elevations (Girardin et al., 2010), potentially driven by greater nutrient availability and temperature, may also stimulate $CH_4$ production in the rhizosphere through the supply of labile substrates to methanogenic communities or maintenance of anaerobic microsites through the $O_2$ demand of heterotrophic respiration (Bodelier, 2011). Such a mechanism, not observed in this data, might be supported by a positive relationship between net $CH_4$ and $CO_2$ fluxes (Verchot et al., 2000)."*

Pg 13, line 10: discussed where? Are you referring to Pg. 11? If so, this is again quite vague. Can you suggest another possible driver?

We have expanded the text on Page 13: *"From these contrasts, it is possible to suggest that relationships with temperature identified here, as discussed with reference to spatial correlations across the Peruvian transect in the previous section, and in Ecuador (Wolf et al., 2012) result from covariance with other drivers like soil moisture or nutrient availability rather than as a result of the temperature sensitivity of $CH_4$ uptake."*

Pg 13, line 17: what mechanism? You comment on differences and go on to suggest they may be related to soil structure, but can you specifically describe a possible mechanism?

We have expanded the text on Page 13: *" Increased WFPS, a function of decreasing soil porosity and increasing precipitation across the transition from upper montane to premontane forests, appears to limit $CH_4$ uptake in Peru through diffusional constraints on the supply of $CH_4$ to methanotrophic communities. Whilst a similar pattern in soil porosity with elevation can be inferred from the presence organic horizons in montane forests studied in Ecuador (Wolf et al., 2012), the alignment between increasing precipitation and decreasing $CH_4$ uptake across this transect might suggest that diffusional constraints, in response to changes in soil moisture, might provide a generalised explanation for the patterns observed. Indeed, Veldkamp et al. (2013) invoke gas diffusional control to explaining positive correlation between net annual CH4 fluxes and rainfall in a meta-analysis of 7 tropical forests above 800 m elevation."*

Figure 4: Double-check your statistical theory here, but as they are currently presented, I don't think these graphs should have regression lines. I believe that once you choose to treat the 3 elevations as forest replicates, the within-elevation replicates would need to be averaged in order to avoid pseudoreplication.

Reviewer 2 also raised this point and we have copied our response here. Reviewer 2 also raised this point and we have copied this response there. Our decision to treat measurement plots within a 'forest type' (or elevation band) as independent replicates of net $CH_4$ exchange is based on the assumption that spatial autocorrelation is limited to short distances (i.e. operating at sub- plot scales of ~ 1 to 10s of m). The plots in our study were more than 100 m apart. We treat our observations as longitudinal data to investigate the possible drivers of the relationship between net $CH_4$ flux and elevation within our study area. In an attempt to synthesise this information, we then discuss $CH_4$ exchange in terms of the ecosystem transitions (or 'forest types') seen across the landscape. This approach is adopted from the literature, for example, across montane forest landscapes (Purbopuspito et al., 2006, p.3) and more recently across lowland tropical forest landscapes (Hassler et al., 2015). We state our approach in the manuscript on Page 6, Line 19 – 21: *"Despite the three plots within each forest type broadly occurring within the same forest stand they were considered independent replicates of forest type as spatial correlations between net $CH_4$ fluxes in tropical forests are small (Ishizuka et al., 2005a; Purbopuspito et al., 2006)"*. However, we acknowledge the concerns of both reviewers. The use of site (n = 3) or plot (n = 9) means in correlation tests does not fundamentally change the pattern or effect size of the relationships which form the basis of our discussion in section 4.2. For example, focussing on the most robust relationships identified between $CH_4$ exchange and environmental conditions (Table 3 (Pearson's r > 0.8 , p < 0.05, n = 9) and then graphed in Figure 4): n = 3, net $CH_4$ flux vs. elevation (Pearson's r = -0.85, p = 0.35), soil temperature (Pearson's r = 0.86, p= 0.34), WFPS (Pearson's r = 0.99, p = 0.10) and NO3 (Pearson's r = 0.92, p = 0.25).
In an attempt to minimise confusion caused by the text, we have altered the somewhat unclear use of 'site' on Page 4 to fall into line with the way we treat the data and how the experimental approach along this transect has previously been described (e.g. Teh et al., 2014, p.2)

Throughout the paper (most mistakes occur in the discussion) use only past tense rather than mixing past and present.

Done, thanks.

When citing references, it is more helpful for the reader if you put references directly behind the information they refer to instead of grouping them at the end of the sentence (i.e. Pg 11, line 5, Pg 12, line 3, Pg 13, line 21).

Done, thanks.

To avoid unnecessary confusion, consider using 'efflux' or 'emissions' to refer specifically to positive fluxes. For example, on Pg. 8, line 3, one could argue that larger fluxes actually occurred at the higher elevations.

Done, thanks.

Pg 8, line 3,16: at lower 'elevations'

Done, thanks.

Pg 8, line 11: use consistent language. Rather than saying 'negative fluxes', use 'uptake' as you do elsewhere.

Done, thanks.

Pg 9, line 2, Pg 10, line 31, Pg 13, line 13: A semi-colon is used to join two related (complete) sentences. Whilst or while are conjunctions, which can be used to compare two (normally contrasting) ideas in a single sentence.

Done, thanks.

Pg 9, line 14: by 'species' do you mean 'forest type'?

Changed to 'compounds'.

Pg 9, line 28: plot 'means'

Done, thanks.

Pg 10, line 2,8: delete 'for example'

Done, thanks.

Pg 10, line 28: delete 'together'

Done, thanks.

Pg 10, line 30: did you do this statistical comparison over time or only comparing wet vs dry season? If the latter is true, then you can't say "with the procession of"

*We have reworded the text on Page 10: " However, the time-series of net CH$_4$ flux (Fig. 2 a) for the upper montane forest does suggest that CH$_4$ uptake increases with the procession of the dry season; ultimately equating to a ~ 30 % difference in uptake between dry and wet season. Our inability to detect a statistically significant difference between seasons at this site (Table 2) may reflect interannual variability, and the fact that environmental conditions change gradually across seasonal transitions (Clark et al., 2014)."*

Pg 11, line 1: 'differences', the fact 'that'

*Done, thanks.*

Pg 11, lines 12-13, Pg 12, line 32: 'source activity' is awkward. Change to 'emissions' or 'efflux'.

*We have reworded the text on Page 11: "Indeed, emissions represented only 1 – 2 % of fluxes in the upper and lower montane forests of this study."*

Pg 11, line 32: 'underpinned' sounds awkward here, perhaps 'supported'?

*Done, thanks.*

Pg 11, line 29-33 (and elsewhere): This part of the discussion would be improved if you indicated which tables/figures correspond to the results you are discussing.

*Agreed, thanks.*

Pg 12, line 17: 'Similar' to

*Done, thanks.*

Pg 12, lines 20, 26: 'the' positive correlations, 'the' wet and dry seasons

*Done, thanks.*

Pg 13, line 1: 'constrained' seems odd in this context, do you mean 'understood'?

*Done, thanks .*

Pg 13, line 4: move the reference to Table 5 from line 1 to here, after 'for these studies'

*Done, thanks.*

Pg 13, line 5: 'in this respect' doesn't fit here, can you reword?

*We have reworded the text on Page 13: "The soils of the montane forests in these three studies are differentiated from those of the premontane forests by the presence of thick organic horizons at the surface."*

References
Hall, S. J., McDowell, W. H. and Silver, W. L.: When wet gets wetter: decoupling of moisture, redox biogeochemistry, and greenhouse gas fluxes in a humid tropical forest soil, Ecosystems, 16(4), 576–589, 2013.

Hassler, E., Corre, M. D., Tjoa, A., Damris, M., Utami, S. R. and Veldkamp, E.: Soil fertility controls soil–atmosphere carbon dioxide and methane fluxes in a tropical landscape converted from lowland forest to rubber and oil palm plantations, Biogeosciences, 12(19), 5831–5852, doi:10.5194/bg-12-5831-2015, 2015.

Purbopuspito, J., Veldkamp, E., Brumme, R. and Murdiyarso, D.: Trace gas fluxes and nitrogen cycling along an elevation sequence of tropical montane forests in Central Sulawesi, Indonesia, Global Biogeochemical Cycles, 20(3), doi:10.1029/2005GB002516, 2006.

Teh, Y. A., Diem, T., Jones, S., Quispe, L. H., Baggs, E., Morley, N., Richards, M., Smith, P. and Meir, P.: Methane and nitrous oxide fluxes across an elevation gradient in the tropical Peruvian Andes, Biogeosciences, 11, 2325–2339, 2014.

---

## Author Response (AR2)

We'd like to thank the editor for suggesting an alternative explanation (copied below in black). We have incorporated this by addition of the following to the discussion on page 14: *"Alternatively, given the lack of temporal correlations between $CH_4$ exchange and soil moisture in these studies, this could also indicate that soil moisture determined at the surface poorly characterised conditions deeper in the profile where greater $CH_4$ oxidation occurred (Purbopuspito et al., 2006)."*

"Dear Authors,

I have read your revised manuscript and welcome the changes that you made based of the reviews of two annonymous reviewers. There is only one thing that I would like you to consider:

on page 15, you suggest that the lack of relationship between net CH4 flux and water content in Ecuador and Indonesia might be caused by the use of gravimetric water content rather than WFPS.

While I understand the reasoning, I don't think that this is the correct explanation since this correlation was also lacking within any of the sites in Ecuador or Indonesia (where a correlation with gravimetric moisture content should appear, if it exists for WFPS, since the conversion from gravimetric moisture content into WFPS would invoke the same particle density and bulk density). I think the more likely explanation is that the soil depth at which the highest CH4 uptake occured (close to the transition from organic to mineral soil), was not the soil depth that was sampled for soil moisture (top 0.05m), and that this explains the lack of correlation between soil moisture and CH4 uptake.

Thank you for submitting a fine study to Biogeosciences.

Best regards,

Edzo Veldkamp"